# Empowering the Blind: Contactless Activity Recognition with Commodity Software-Defined Radio and Ultra-High-Frequency Radio Frequency Identification

**DOI:** 10.3390/s24113645

**Published:** 2024-06-04

**Authors:** Muhammad Zakir Khan, Turke Althobaiti, Muhannad Almutiry, Naeem Ramzan

**Affiliations:** 1James Watt School of Engineering, University of Glasgow, Glasgow G12 8QQ, UK; m.khan.6@research.gla.ac.uk; 2Department of Computer Science, Faculty of Science, Northern Border University, Arar 73222, Saudi Arabia; turke.althobaiti@nbu.edu.sa; 3Electrical Engineering Department, Northern Border University, Arar 73222, Saudi Arabia; muhannad.almutiry@nbu.edu.sa; 4School of Computing, Engineering and Physical Sciences, University of the West of Scotland, Paisley PA1 2BE, UK

**Keywords:** contactless activity monitoring, SDR-RFID, blind, visual impairment

## Abstract

This study presents a novel computational radio frequency identification (RFID) system designed specifically for assisting blind individuals, utilising software-defined radio (SDR) with coherent detection. The system employs battery-less ultra-high-frequency (UHF) tag arrays in Gen2 RFID systems, enhancing the transmission of sensed information beyond standard identification bits. Our method uses an SDR reader to efficiently manage multiple tags with Gen2 preambles implemented on a single transceiver card. The results highlight the system’s real-time capability to detect movements and direction of walking within a four-meter range, indicating significant advances in contactless activity monitoring. This system not only handles the complexities of multiple tag scenarios but also delineates the influence of system parameters on RFID operational efficiency. This study contributes to assistive technology, provides a platform for future advancements aimed at addressing contemporary limitations in pseudo-localisation, and offers a practical, affordable assistance system for blind individuals.

## 1. Introduction

According to the world health organisation, more than 2.2 billion people globally are affected by blindness and vision impairment, with about 75% of these cases occurring in individuals over the age of 50 [1,2,3]. This issue is growing, especially as the global population is expected to reach 11 billion by 2100, which could result in up to 0.6 billion people suffering from visual impairments [2]. In the UK, around 2.28 million people suffer from moderate to severe vision impairment, and about 0.17 million are completely blind [4]. These conditions not only affect quality of life but also impose substantial economic burdens, with global costs estimated at GBP 337 billion annually [1]. The prevalence of vision impairment, particularly among the elderly, increases the risk of accidents, loneliness, and early admission into nursing homes. These challenges highlight the need for innovative, contactless monitoring and support systems to improve the lives of those with visual impairments and effectively address the diverse challenges they face [1,4].

Assistive technologies are essential for visually impaired individuals, enhancing their perception of the environment and interaction capabilities. While accessible for their simplicity and cost-effectiveness, traditional aids, including the white cane, offer limited environmental awareness [5]. However, advancements in electronic travel aids (ETAs) like QD laser [6] and Aira [7] have been proposed, but they do not fully address all human factors [8]. Furthermore, recent advancements involving optical and ultrasonic sensors [9,10] show promise but face challenges like varying light conditions, privacy concerns [5,11], and difficulties with reflective surfaces [12]. These limitations underscore the need for a new, user-friendly, intuitive, reliable, and low-cost contactless system that ensures safety and an effective response, thus addressing the practical requirements for assistive devices for the visually impaired.

This study presents an innovative approach to detecting the movements of blind individuals by integrating radio frequency identification (RFID) technology with a software-defined radio (SDR) system. The primary motivation behind this study is to develop an RFID-based system capable of monitoring blind individuals’ movements in a contactless manner. This system utilises battery-less UHF tag arrays and a complete SDR reader with coherent detection capabilities. By exploiting the already present preambles in Gen2 RFID systems, our approach avoids rate degradation and enhances the efficiency and responsiveness of the system in real-time environments. This method utilises a fully coherent, full-duplex Gen2 reader [13], distinct from traditional SDR systems, for its coherent, linear processing capabilities. It features a single transceiver board based on the USRP2 (N200) platform, utilised to detect blind individuals in a contactless manner through the RFID tag received signal strength (RSSI). The technique uses memory effects from FM0 line coding in RFID tags and uses Gen2 preambles for effective channel estimation, ensuring data rate maintenance. The system comprises an SDR-based modular UHF RFID reader and a set of modular, passive, and computational UHF RFID tags. This setup, following software architecture principles [13,14], works with standard commercial RFID tag sensors under the Gen2 protocol [15].

The specific contributions are as follows:1.Developing a fully integrated SDR-RFID system that uses Gen2 preambles to enhance signal processing efficiency without suffering from rate degradation.2.Demonstrating the system’s capacity to use a battery-less UHF RFID tag array to accurately detect and monitor the mobility of blind individuals in indoor environments.3.Highlighting the technology’s application in healthcare environments, where its real-time, contactless monitoring can substantially improve the safety and mobility of visually impaired individuals.

This configuration is not only cost-effective but also an invaluable resource for further research, particularly in the field of developing assistive technologies for individuals with blindness or visual impairments.

## 2. Related Work

Software-defined radio (SDR) readers have advanced significantly in the last decade, particularly in the area of Gen2 RFID technology. Early proposed systems utilised Miller line codes and non-coherent reception, requiring separate boards for signal transmission and receiving [14,16]. These systems have improved as listener-only systems (sniffers) utilising non-linear processing have been developed [17,18]. A significant advancement that enhanced the evaluation of RFID tag performance was the replacement of Buettner’s system with a solitary transceiver board [14,19]. Despite these advances, the efficiency and versatility of the early systems were restricted by the USRP1 SDR and outdated GNU radio software. Recently, RFID systems have been upgraded with new hardware and software such as GNU Radio v3.6 and, subsequently, GNU Radio v3.7 with USRP2 SDR, mostly for commercial tags [20,21]. However, their application and flexibility for research are limited, since these modifications are rarely made publicly available [21]. Furthermore, research has been conducted on customised FPGA-based systems [22,23]. Developing on the RFID monitors developed in [14,16], which were the initial steps towards full-protocol interrogators, a complete SDR implementation of an EPC-C1G2 reader is presented in [14]. This implementation involved a USRP 1 with a RFX900 daughterboard’s RF frontend, separate $R_{x}$ and $T_{x}$ antenna, and implementation using the open-source GNU Radio software, marking substantial progress in flexible RFID reader development.

The development of a fully coherent EPC-C1G2 SDR reader [13], building on [14], marked a significant advancement in RFID technology. This open-source design, which is compatible with commercial UHF RFID tags employing FM0 encoding and a 40 kHz backscatter link frequency (BLF), was a notable leap forward. The reader utilized a single transceiver board integrated into a USRP 2, equipped with separate $R_{x}$ and $T_{x}$ antennas. However, this configuration did not support the investigation of common commercial scenarios that usually involve a singular $R_{x}$/$T_{x}$ antenna. This limitation highlights the continued need for more adaptable solutions in RFID technology. Specifically, for research applications, the ability to modify parameters of the EPC-C1G2 or to adapt the protocol remains a critical requirement.

## 3. Materials and Methods

In uplink RFID communication, where information flows from the tag to the reader, the process begins with the reader transmitting a carrier wave (CW). The RFID tag then modulates this wave to transmit data by switching its antenna load between two distinct states, effectively using binary modulation. This modulation changes the wave’s reflection coefficient, imprinting the tag’s information onto it. The reader receives a signal that is a composite of its original transmission and the tag’s backscattered signal. The complexities of this received signal at the reader’s end are accurately encapsulated by equations given by [24], offering a detailed mathematical representation of this process in Equation (Equation 1).
(1)$$y\left(t\right)=\left[M_{c}+M_{m}\;x\left(t\right)\right]e^{+j2\pi ft}+N\left(t\right)$$

The received signal at the reader is denoted by y(t). The signal comprises several components. Firstly, $M_{c}\in C$ represents the DC component, representing the unmodulated backscatter from the RFID tag and the constant part of the CW. The second component, $M_{m}\in C$, is the modulation component, which is affected by various factors such as link coefficients between the reader’s transmitting antenna and the tag, tag antenna characteristics, scattering efficiency, and the transmitted carrier’s power. Another important element is x(t)—the binary real-valued waveform backscattered by the tag. Additionally, the system accounts for the carrier frequency offset (CFO), denoted by *f*, which is zero (Δf=0) in this setup due to the utilisation of the same oscillation signal for both transmission and reception. Lastly, N(t) represents the complex Gaussian noise in the system, primarily originating from the receiver.

### 3.1. Experimental Setup

The experiment of this study was carried out in a 25 m^2^ room in the James Watts South building at the University of Glasgow, with ethical approval from the University’s Research Ethics Committee (approval nos.: 300200232, 300190109). Its primary aim was to assess the readability of RFID tags over various distances. For this, an array of five EPC Class 1 Gen 2 standard tags, named the BlindDetectorBoard, was attached to a paper board. Readability tests were conducted at distances of 1, 3, 5, and 7 m from the reader antennas, as detailed in Figure 1. The experiment was designed to be efficient, matching the number of slots in each round to the number of tags to optimize the testing process.

For experimental purposes, two circularly polarised antennas were used, with gains of 20 dB for receiving ($G_{R_{x}}=20$ dB) and 30 dB for transmitting ($G_{T_{x}}=30$ dB). These antennas were strategically positioned horizontally from the centre of the BlindDetectorBoard, and the subject in the experiment was positioned 10 cm away from this board ($D_{BS}=10$ cm). It is important to note that the data collection involving the subject’s activities was conducted in a line-of-sight (LoS) setup, where the direct path between the RFID reader, the subject, and the tag array board was unobstructed, allowing clear data collection on movement and activity positioning against each tag. In the NLoS setup, elements such as furniture and metallic boxes were placed to create a rich environment that mimicked typical indoor scenarios with potential signal obstructions, although no subject activities were performed in this setup. The aim was to test the system’s robustness under conditions where a direct line of sight is not maintained. The reader’s transmission settings were adjusted to ensure that each tag responded in every inventory round, thereby facilitating comprehensive data collection. This approach resulted in a total of 60 inventory rounds ($N_{rounds}=60$), creating 5×60=300 slots ($N_{slots}=300$). Furthermore, the experiments also involved varying the transmission power levels (30/30/30/40/30/30 dBm) while adjusting the receiver power (20/20/20/20/30/40 dBm). This variation demonstrated that increasing the receiver gain enhanced the reader’s performance, moving it closer to the efficiency of an ideal reader. Notably, the read rate exceeded expectations due to the capture effect, where slots containing collided tag signals could be decoded if there was a significant power difference between them ($\Delta P>P_{threshold}$). The data collection process involved three main components: the UHF RFID reader, hardware setup, and the software setup, which are detailed in the subsequent subsections.

#### 3.1.1. UHF RFID Reader

The Gen2 reader, structured according to the six-block design documented in [14], incorporates USRP source, Matched Filter, Gate, Tag Decoder, Gen2 Logic, and USRP sink blocks, a configuration illustrated in Figure 2 [13]. Each of these blocks is linked to a buffer, known as File sink, which captures and stores output for debugging purposes. In addition to processing incoming data, the Gen2 reader is responsible for transmitting information to the tags. This reader operates on the principle of framed slotted aloha (FSA) for accessing multiple tags simultaneously (see Figure 3b), where the probability of a successful transmission, $P_{\mathrm{success}}=Np{\left(1-p\right)}^{N-1}$ with $p=\frac{1}{S}$, details the operational mechanics behind the simultaneous tag access.

An inventory round begins when the Gen2 reader sends a Query command, establishing critical communication parameters such as the tag rate, data encoding (FM0 or Miller), and the number of slots, as shown in Figure 3a. In this phase, each tag randomly selects a number between 1 and N (N being the total number of slots) and sends a 16-bit random sequence (RN16) in its chosen slot, preceded by a preamble for identification. The reader acknowledges a tag with an ACK command and receives a 135-bit response containing the tag’s EPC (a 96-bit identifier), CRC, and other bits. Notably, the reader must reply within 500 µs, especially at the minimum supported tag rate of 40 KHz, to maintain efficient communication in the system.

#### 3.1.2. Hardware Setup

In this study, we utilised a flexible SDR system, specifically the Ettus Research USRP N210 (see Figure 4a) combined with an RFX900 transceiver, capable of operating in the 865–868 MHz range. In order to achieve good performance in the UHF spectrum (865–960 MHz), zebra UHF RFID tags were utilised, as shown in Figure 4c; the tags have an integrated circuit for data processing and an antenna for signal transmission, which are essential for modulating and demodulating RF signals and for memory management and backscatter communication. This setup enabled full duplex communication, allowing for simultaneous signal transmission and reception. System control was managed through a laptop running Ubuntu 16.04, connected to the USRP through a gigabit network cable. Building on the open-source framework mentioned in [13], we expanded our system to fully support the read cycle according to the EPC C1G2 communication protocol. This enhancement is particularly beneficial for capturing data from RFID sensors, as it allows for the full execution of the reader-tag communication sequence, including ReqRN-Handle-Read-Data commands. To integrate with our tag array, we adapted the reader to accommodate backscatter link frequencies of 40 kHz and 160 kHz. Furthermore, we connected the reader with a circularly polarized antenna (250×250×14 mm) with an 8.0 dBi gain, ensuring robust and effective communication within the RFID system.

#### 3.1.3. Software Setup

In this study, the SDR-RFID system is executed on a laptop with Ubuntu 16.04 using GNU-Radio 3.7.4. This system’s architecture is based on six key blocks: USRP source, matched filter, gate, tag decoder, Gen2 logic, and USRP sink, as shown in Figure 2. These blocks collectively handle the critical tasks of receiving and transmitting samples through USRP. To enhance the interaction with computational RFID tags array, we have integrated custom C++ modules with Python scripting, drawing inspiration from the open-source framework in [13].

Block Functionality and Signal Processing
Matched filter block: Its primary function is to filter the received signals using a square pulse of length L/2=25. To optimize efficiency, the signal is downsampled by a factor of 5, reducing computational load.Gate block: This block is responsible for identifying reader queries, an essential function given the reader’s full-duplex capability, which involves simultaneous transmission and reception. By monitoring signal amplitude, it detects transmitted commands and selectively processes samples corresponding to the tag’s responses, ensuring accurate and efficient signal processing.Synchronisation and Decoding
Tag decoder block: This block is responsible for managing frame synchronisation, channel estimation, and detecting tag responses. It achieves synchronisation by correlating the received signal with a known preamble, focusing on RN16 and tag ID (EPC) sequences.Handling Bit Duration Variations: A common issue in RFID systems is the fluctuation in the tag’s nominal bit duration, which can be altered by up to 22%. Our reader addresses this by operating at lower data rates (40 KHz), reducing the impact of these variations.Synchronisation Strategy: Although symbol-level synchronisation errors are noted, particularly when decoding longer sequences like the tag ID and CRC, these issues are absent in the shorter RN16 sequence. To address these synchronisation issues, we utilise the initial sampling instant $\tau_{opt}$ by correlating with the preamble. Subsequently, the optimal symbol rate $S_{opt}$ is selected, which maximises signal energy, leading to more accurate sampling at each half symbol period. This process is represented in Equation (Equation 2):
$$S_{opt}=\max_{S}F\left(S\right)$$
(2)$$F\left(S\right)=\sum_{k=0}^{2(M-1)}{\left|y_{filtered}\left[\tau_{opt}+k\frac{S}{2}\right]\right|}^{2},$$
where F(S) is the function to maximise, representing the sum of the squared magnitudes of the filtered signal $y_{filtered}$ sampled at intervals determined by *S*. $\tau_{opt}$ is the optimal delay that maximises the signal’s strength, *M* represents the total number of intervals considered for sampling within the signal’s duration, and *S* is the sampling interval variable over which the maximisation is performed. The $k\frac{S}{2}$ adjusts the sampling points within the signal, aiming to find the sampling rate that maximises the energy of the sampled signal.Command Generation and Efficiency
USRP Sink Block: Following tag data decoding, this block generates and transmits appropriate commands (Query, ACK, QueryRep) based on the output from the tag decoder block with (ReqRN, Handle, Read-Data).DAC rate: The digital-to-analog converter rate is set at 1 MS/s to maintain efficient operation.

### 3.2. Encoding and Signal Analysis

Our study focuses on the encoding techniques of RFID tags, particularly examining FM0 and Miller (M∈{2,4,8}) encoding. In FM0 encoding, every bit boundary exhibits a level transition, and ‘0’ bits have an additional mid-bit transition, creating four potential waveforms per bit (see Figure 5) with dashed lines separating bit boundaries. However, as identified in [17], shifting the analysis of the waveform by half a bit period before the start of a bit simplifies these waveforms to two distinct pulse shapes.

## 4. Multiple Tag Model

In this section, we extend our study to the communication between the reader and multiple tags. When multiple tags respond simultaneously, collisions occur. For a scenario involving multiple tags, the received digitised signal, after DC offset estimation and removal, is represented in Equations (Equation 3)–(Equation 6):(3)$$y_{\mathrm{signal}}\left[m\right]=\sum_{i=1}^{N_{\mathrm{tags}}}h_{i}x_{i}\left[m\right]$$
where $y_{\mathrm{signal}}\left[m\right]$ is the combined signal received from all $N_{\mathrm{tags}}$, where each tag’s signal $x_{i}\left[m\right]$ is modified by the channel’s impact $h_{i}$.
(4)$$y\left[m\right]=y_{\mathrm{signal}}\left[m\right]+n\left[m\right]$$

y[m] represents the total received signal, which includes both the combined response from all tags, $y_{\mathrm{signal}}\left[m\right]$, and the noise n[m]:(5)$$x_{i}\left[m\right]=\sum_{p=0}^{P}S_{d(p)}^{i}\left[m-pK-\tau_{i}\right]$$

In these equations, $S_{d(p)}^{i}$ is the p+1th transmitted pulse for tag *i*, $K=T/T_{s}$ is the oversampling factor, and $n\left[m\right]∼\mathrm{CN}\left(0,2\sigma_{n}^{2}\right)$.

The tags are synchronised ($\tau_{i}=0$ for all *i*) and using the same nominal bit duration, the signal after synchronisation and matched filtering can be expressed as follows:(6)$$y=\sum_{i=1}^{N_{tags}}h_{i}^{\prime }x_{i}+n,$$

Here, $x_{i}\in \left\{e_{0},e_{1}\right\}$ represents the state vectors for each tag, and $n∼\mathrm{CN}(0,K\sigma_{n}^{2}I_{2})$.

The signal-to-noise ratio (SNR) in this multi-tag scenario is defined as follows:(7)$$H=\sum_{i=1}^{N_{\mathrm{tags}}}{\left|h_{i}^{\prime }\right|}^{2}$$
(8)$$\mathrm{SNR}=\frac{\sum_{i=1}^{N_{tags}}{\left|h_{i}^{\prime }\right|}^{2}}{K\sigma_{n}^{2}},$$

This model captures the complexity of interactions when multiple tags are involved, showcasing the reader’s ability to process and distinguish signals from various sources simultaneously.

## 5. Experimental Results

In this study, we conducted different experiments to evaluate the reading rate and range performance of an RFID reader and two circularly polarised antennas boasting an 8 dBic gain. We assessed two detection methods and a collision recovery algorithm through practical comparison.

### 5.1. Performance of RFID Reader Detection Methods

In our experiment, we placed a tag at varying distances from 1 to 4.5 m away from the reader. With the setup involving 1 slot and 500 queries ($N_{\mathrm{queries}}=500$), we conducted tests using both coherent and noncoherent detection methods under different transmission power levels. The results, illustrated in Figure 6a, allowed us to calculate the success ratio (SR), which is the proportion of correctly decoded EPC messages ($N_{\mathrm{correct}}$) to the total number of queries ($N_{\mathrm{queries}}$), expressed as $SR=\frac{N_{\mathrm{correct}}}{N_{\mathrm{queries}}}$. Notably, coherent detection exhibited an 18% higher EPC decoding success rate (ΔSR=18%) at lower transmission powers. However, as transmission power was increased ($P_{\mathrm{transmission}}$), the difference in performance between the two detection methods decreased.

### 5.2. RFID Reader Range Measurement

In our experimental study, we evaluated the read range of an RFID reader by adjusting the receiver gain to $G_{R_{x}}=20$ dB and the transmit power to $P_{T_{x}}=30$ dBm. The UHF RFID tags array were positioned at distances ranging from 1 to 7 m and issued 500–1500 queries per tag. We observed a success ratio of nearly 100% up to 6 m for Tag A, and 4 m for multiple tags, as illustrated in Figure 6b. Specifically, to detect walking movements and directions within a 2.5 m range, we aimed to perform activity identification for blind individuals by detecting tag reading 12 times within a one-second interval. This was achieved by monitoring RSSI reading blockages, where fewer blockages indicated the movement of a blind individual within the area. The detected movements are visually represented in green, while no movement detection is depicted in blue, as illustrated in Figure 7.

To provide further clarity regarding walking direction and the specific location of activities, we continuously recorded the RSSI data at five-second intervals. We labeled five tags from 0 to 4 for these experiments to facilitate precise localisation. In Figure 7a, the subject began walking from Tag 0 towards Tag 4. Tags 1 and 4, highlighted in green, indicated active movement detection (tag reading blocked shows green), signaling the subject’s progressive movement from the start to the end of the array. In contrast, Figure 7b uses tag blockage to indicate both the detection and the specific location of activity during movement. The RSSI variations can easily recognise each activity at the same location, with tags being blocked causing a drop in the RSSI values. Notably, the threshold for RSSI strengths was determined by observing the maximum and minimum values, which were recorded at −55 dbm and −69 dbm, respectively. The latter value, −69 dbm, was selected as the threshold, represented with red dots, taking into account potential instances of non-reading or blocking of tag activity detection. This visualisation and method highlight the system’s capability to monitor and recognise the specific movement or walking directions of blind individuals, showcasing significant potential for enhancing assistive technologies for the visually impaired.

### 5.3. Technical Analysis of RFID Reading Rates

In our study, a significant focus was on analysing the RFID system’s reading rate, essential for applications like activity recognition for the blind. We calculated the number of successful EPC reads per second, considering the total cycle time for one EPC read. This cycle time includes key intervals:Total Cycle Time for One EPC Read: Comprising reader to tag response (T1_D = 240 µs), tag response to reader transmission (T2_D = 480 µs), RN16 transmission (RN16_D), and EPC transmission (EPC_D) times.RN16 Transmission Time (RN16_D): Computed as (17+6) × TAG_BIT_D, with TAG_BIT_D being the duration of one tag bit in microseconds.EPC Transmission Time (EPC_D): Calculated using EPC_BITS=129.
TotalCycleTime=T1_D+T2_D+RN16_D+EPC_D

RN16_D is calculated as 575 microseconds, and EPC_D as 3375 microseconds, leading to a total cycle time of 4670 microseconds for one EPC read. Consequently, the system can perform approximately 214 reads per second at a one meter distance, an important factor in real-time monitoring and detecting activities of blind individuals.

### 5.4. Evaluation of RFID System in Two Distinct Scenarios

This section evaluates the performance of an RFID system under two scenarios, focusing on the impact of inventory rounds, slot count, and query structure on the efficiency of EPC message decoding. Key to each scenarios is the Query command, which sets essential parameters for the inventory round, including select (SL) and inventoried flags influencing tag participation. It also establishes the slot counter Q (values from 0 to 15), which determines the number of slots $\left(N=2^{Q}\right)$ in each round, as illustrated in Table 1 and Table 2. Tags respond with an RN16 message when they meet the slot counter requirements. Furthermore, the Query command adjusts the backscatter link frequency (BLF) and encoding, both vital for effective data transmission and decoding.

#### 5.4.1. Evaluating RFID System Efficiency under Variable Inventory Configurations (Scenario 1)

In the first scenario, the RFID system’s performance in decoding EPC messages under different conditions was assessed by varying its configuration throughout two distinct sets of inventory rounds. In the first round of inventory, the slot counter *Q* was altered, resulting in 32 rounds of inventory and 500 total queries. A 12% success ratio ($\frac{60}{500}=0.12$ or 12%) was achieved with 60 successfully decoded EPC messages due to this alternate configuration. Since they receive signals more strongly, the tags in this phase allow improved read success. In contrast, the second set utilised a different configuration that significantly increased the number of inventory rounds to 251 for the same number of 500 queries. This adjustment aimed to explore the system’s behaviour under a high-density round environment but resulted in a reduced number of successfully decoded messages at 45, which translates to a lower success ratio of 9% ($\frac{45}{500}=0.09$ or 9%). The extended number of rounds increased the potential for collisions or interference, affecting the decoding success. This controlled variation highlights the impact of inventory round density on the efficiency of tag reading and EPC message decoding, demonstrating the trade-offs between inventory round settings and successful tag interrogation.

#### 5.4.2. Balancing Efficiency and Collision Risks through Slot Counter Optimisation (Scenario 2)

In the second scenario, the slot counter *Q* was set to balance efficiency and collision probability. In the first inventory round setup, *Q* was likely set higher, resulting in 16 slots per round ($N=2^{4.5}$), which spanned 32 inventory rounds accommodating 500 queries. This configuration aimed at reducing the number of rounds but increased the potential for collisions due to more slots per round, achieving a success ratio of 5% ($\frac{25}{500}=0.05$ or 5%). The tags were positioned further from the reader, up to 4 m, which might have contributed to the lower success rate due to decreased signal strength. However, the second setup involved a drastic reduction in the number of queries to 55, yet the inventory rounds were closely numbered at 28, indicating a much lower *Q* value, which significantly increased the efficiency of tag reads. This strategic adjustment led to an improved corrected decoded EPC (CD-EPC), suggesting that the proximity or responsiveness of these tags was better optimised in this round. The distribution and read success in this configuration underscored the critical balance between slot count and operational distance, demonstrating a notable increase in decoding success to approximately 36% ($\frac{20}{55}=0.364$ or 36.4%).

## 6. Conclusions

This study has successfully implemented an RFID-based detection system integrating SDR and RFID technologies, tailored specifically to enhance the mobility and safety of blind individuals. The system detects movements in a contactless manner using a tag array, which simplifies interactions compared to traditional tangible-based aids and reduces the cumbersome nature of direct physical contact. By leveraging pseudo-localisation, the system effectively monitors and determines the walking directions of individuals, thereby significantly improving navigation in indoor environments. Futhermore, the system highlights a significant advancement in assistive technologies by demonstrating the ability to identify individuals within a 4-m range using UHF Passive RFID tags. Additionally, our findings detail the complexities involved in managing multiple tag scenarios and illustrate the influence of system parameters on the efficiency of RFID technology. This research marks an important step toward sophisticated, contactless assistive solutions that significantly enhance environmental awareness for blind individuals, paving the way for further developments in this area.

## Figures and Tables

**Figure 1 sensors-24-03645-f001:**
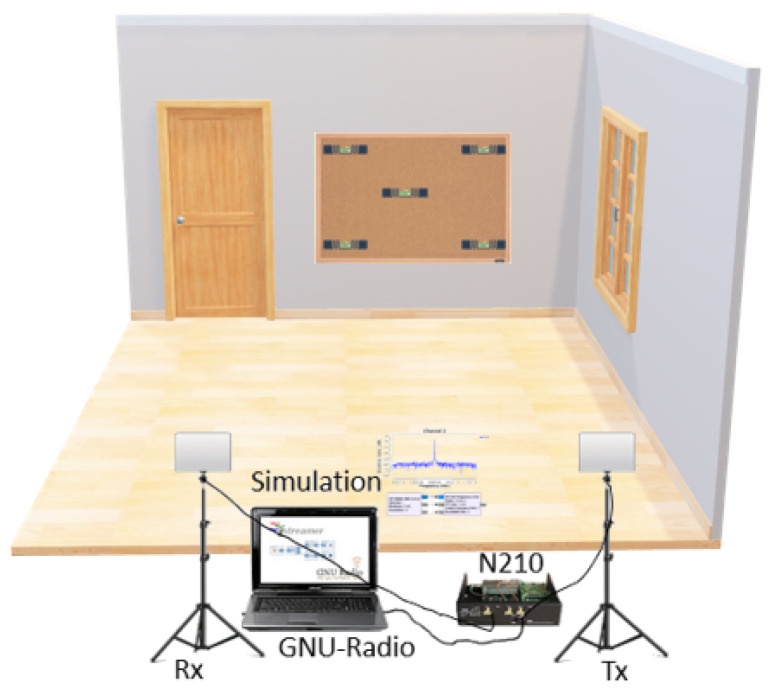
Experimental setup for blind individual detection.

**Figure 2 sensors-24-03645-f002:**
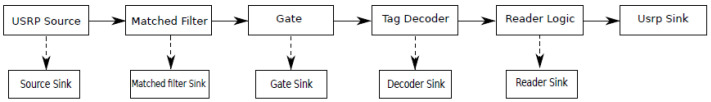
Architecture of the SDR Reader.

**Figure 3 sensors-24-03645-f003:**
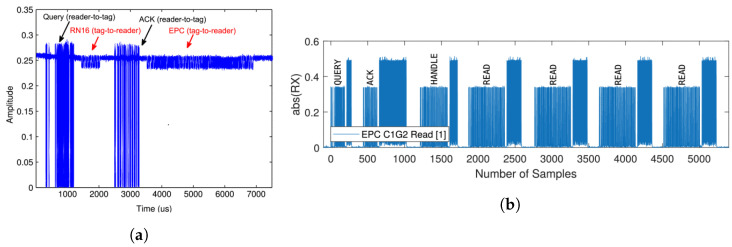
Bidirectional communication read-tag frame and EPC-C1G2 protocol exhibiting continuous multi-read. (**a**) Read-tag frame [13]. (**b**) Multi-data read [25].

**Figure 4 sensors-24-03645-f004:**
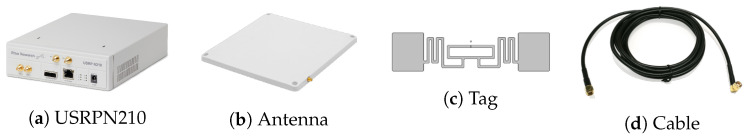
Components of the experimental setup for SDR-RFID-based blind detection.

**Figure 5 sensors-24-03645-f005:**
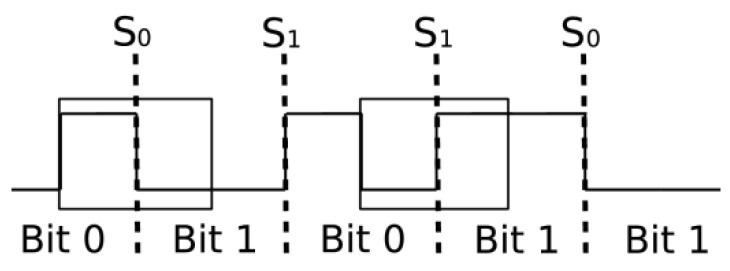
FM0 encoded signal.

**Figure 6 sensors-24-03645-f006:**
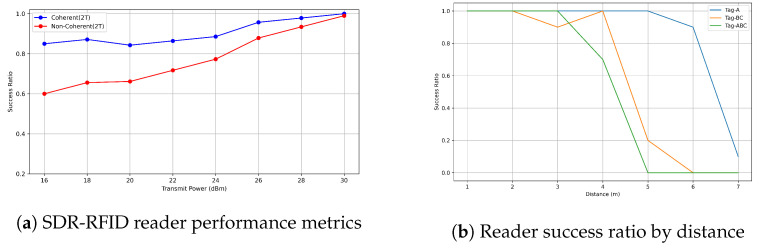
RFID reader evaluation: (**a**) Detailed performance metrics for activity monitoring and (**b**) success ratio by distance illustrating range effectiveness.

**Figure 7 sensors-24-03645-f007:**
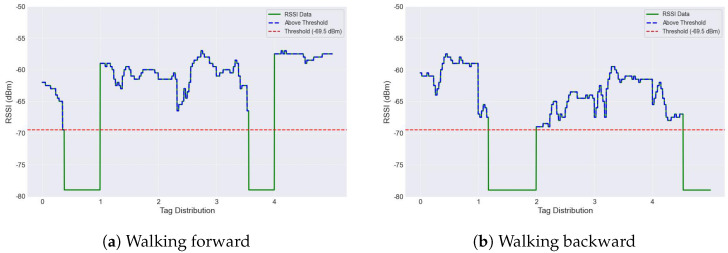
Diagram depicting the recognition of walking directions using RSSI data.

**Table 1 sensors-24-03645-t001:** Performance results for Scenario 1 (six tags) showing the number of queries sent, corrected decoded EPC (CD-EPC), and tag reads over two inventory rounds (IR).

Inventory Details			Tag IDs in Hexadecimal					
**Queries**	**IR**	**CD-EPC**	**00**	**4b**	**6b**	**72**	**af**	**d9**
500	32	60	16 reads	10 reads	9 reads	15 reads	8 reads	2 reads
500	251	45	14 reads	11 reads	6 reads	4 reads	7 reads	3 reads

**Table 2 sensors-24-03645-t002:** Performance results for Scenario 2 (three tags) showing the number of queries sent, corrected decoded EPC (CD-EPC), and tag reads over two inventory rounds (IR).

Inventory Details			Tag IDs in Hexadecimal		
**Queries**	**IR**	**CD-EPC**	**00**	**4b**	**6b**
500	32	25	19 reads	3 reads	3 reads
55	28	20	2 reads	12 reads	16 reads

## Data Availability

The source code for the implementations discussed in this paper is available on GitHub https://github.com/ZakirAfridi/Gen2-UHF-RFID-Reader-For-BlindPeople-Detection/ (accessed on 23 May 2024) a repository to ensure reproducibility and facilitate further research.

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
