# Peer review of "Empowering the Blind: Contactless Activity Recognition with Commodity Software-Defined Radio and Ultra-High-Frequency Radio Frequency Identification"

_sensors, 2024, doi:10.3390/s24113645_

Round 1
Reviewer 1 Report
Comments and Suggestions for Authors
This paper aims to integrate RFID technology with a software-defined radio system to contactlessly recognize the activities of the blind individuals in healthcare environments by managing multiple tags. Overall, the paper is well-structured and clearly presented. Below are some recommendations and/or comments from the reviewer if the authors would like to make some improvements.
Major comments:
1. Abstract section: The poor logic flow in this section hinders the reader’s willingness to continue reading the entire paper despite the subsequent sections being well-written. Thus, it is recommended to rewrite this section by sequentially considering the research motivation, research problem, proposed method(s), highlights of results, key indication(s) of the results, and the contribution of your research to the domain of knowledge.
2. For application, the paper targeted activity recognition of the blind. However, there is no information about how the activities of the blind individuals are recognized. Based on the signals from the RFID tags, did the authors develop machine learning models to do the activity recognition? Or did the authors manually recognize them? In addition, what types of activities were recognized? If none, the application scenario would better fit the pure detection of blind individuals to capture their occupancy within a defined distance range using the system developed by the authors. As a result, the authors may want to revise the narratives about the application scenario for the developed system across the entire paper.
Minor comments:
1. Line 2: What perspectives of the blind people are monitored?
2. Line 31: Typing issue with “ad- dress”
3. Lines 115 – 116: How do subject’s activities and the surrounding environment influence the data collection?
4. Texts in Figures 2, 3, and 7 are barely readable
5. Lines 302-303: How can the proposed system assist blind individuals?
Reviewer 2 Report
Comments and Suggestions for Authors
This paper presents a computational radio frequency identification (RFID) system based on Software Defined Radio (SDR), which uses coherent detection to monitor blind people. Overall, the paper is interesting, and the results are convincing. Before possible publication, the following concerns should be addressed.
1. Line 17 : [1?, 2] should be [1, 2], right?
2. Line 22 : "billion annually [?]", the reference is missing.
3. Line 26 : [3?] should be [3], right?
4. The authors introduced so many previous work. So, what's the motivation of the proposed method? The authors should clarify the motivation in Introduction.
5. The reviewer suggests the authors conclude the main contributions of the paper in Introduction.
6. Related work section, related work should not only introduce previous work, but also analyze these work and give some comments.
7. Line 90: "in Equation (1)" should be "in (1)".
8. The resolution of Figure 1 should be enhanced.
9. Figure 7 is not clear, the reviewer suggests the authors arrange the three subgraphs in three rows.
10. Conclusion should conclude the main findings from previous experimental results.
Reviewer 3 Report
Comments and Suggestions for Authors
Dear Authors,
Your work seems interesting, and you have tried to provide valuable insights into implementing and evaluating the RFID system. However, there are some suggestions to enhance the quality of the research.
-The abstract needs to be revised with the novel contribution.
-The introduction section needs to be revised.
-Do a more in-depth study of the existing literature and provide evidence of the robustness and difference, or compare your implementation and evaluation to the existing ones.
- More studies need to be done on the co-existing system.
- Provide the architecture of the tag.
- There is a range limitation; conditions in healthcare facilities or outdoor settings may require a more extended range.
- Localization must be addressed seriously in assistive technologies for the visually impaired to enhance their quality of life. Relying on the system estimates may introduce inaccuracies in the dynamic environment and when the individuals are in motion.
-This research lacks the practicality to enhance assistive technology, focusing only on identification.
- Managing multiple tags and readers can be challenging in a dense population. There may be complexities in synchronization, interference management, and overall performance.
-The experiment needs to be run one more time.
- Conclusions need to be rewritten.
Round 2
Reviewer 2 Report
Comments and Suggestions for Authors
The authors have addressed my concerns, and I have no more comments.
Author Response
Dear Reviewer, Thank you so much for your valuable insight and time to review my paper for better quality after incorporating all your comments.
Reviewer 3 Report
Comments and Suggestions for Authors
Dear Authors,
Thank you for revising your work. Please check the typological errors like "illustrated in Table ?? and 2. " Line number 318, in your manuscript.
Author Response
Thank you for mentioning typological errors, and I have incorporated those typological errors.